# Synthesis and Novel Purification Process of PANI and PANI/AgNPs Composite

**DOI:** 10.3390/molecules24081621

**Published:** 2019-04-24

**Authors:** María L. Mota, Amanda Carrillo, Ana J. Verdugo, Amelia Olivas, Jorge M. Guerrero, Edna C. De la Cruz, Natalia Noriega Ramírez

**Affiliations:** 1CONACYT-Institute of Engineering and Technology, Autonomous University of Ciudad Juárez, Av. Del Charro 610, Ciudad Juárez, CHIH, C.P. 32310, Mexico; 2Institute of Engineering and Technology, Autonomous University of Ciudad Juárez, Av. Del Charro 610, Ciudad Juárez, CHIH, C.P. 32310, Mexico; amanda.carrillo@uacj.mx (A.C.); al183008@alumnos.uacj.mx (A.J.V.); al164312@alumnos.uacj.mx (N.N.R.); 3Center of Nanoscience and Nanotechnology-UNAM, BC, C.P. 22860, Mexico; aolivas@cnyn.unam.mx; 4Center of Advanced Materials Research, S.C., Alianza Norte 202, Research and Technological Innovation Park, Apodaca, NL, C.P. 66600, Mexico; jorge.guerrero@cimav.edu.mx; 5CONACYT-Center of Research in Applied Science and Advanced Technology, Altamira Unit, Tamaulipas, C.P. 89600, Mexico; ecdelacruzte@conacyt.mx

**Keywords:** PANI, PANI/AgNPs, composite, purification

## Abstract

In this work, polyaniline (PANI) is synthesized via oxidative polymerization of aniline and purified using organic solvents where the emeraldine phase is isolated by employing a phase separation system. The above contributes to the increase in the percentage yield compared to previous works and the possibility of being used as a single phase. In addition, the PANI/AgNPs composite is prepared in situ at the polymerization of aniline, adding silver nitrate and glycine to create the AgNPs inside the PANI matrix by controlling the pH, temperature, time of reaction and incorporating a new purification technique.

## 1. Introduction

Polyaniline (PANI), a derivative of polyphenylene vinylene (PPV), belongs to a class of materials known as conducting polymers, combines the electronic and optical properties of some organic semiconductors and metals with the processing advantages of polymers [1]. PANI has a variety of phases with unique and special features such as stability under environmental conditions [2], chemical or electrochemical redox reversibility, high conductivity [3,4], ease of synthesis [5,6], and low-cost in comparison to other conducting polymers, which make it suitable for its application in sensors [7], photochemical cells and anticorrosion layers [8]. Furthermore, PANI has many promising applications in biomedical areas, such as tissue engineering, biosensors and biomedicine. However, in order to improve the thermal and optical properties, the development of composites with the incorporation of several nanomaterials has been studied. For instance, PANI–metal nanoparticles composites have shown enhanced sensing capabilities compared to those of pure PANI, due to the ability of the metal nanoparticles to act as conductive junctions within their chains [9,10]. Its synthesis is carried out by chemical oxidative polymerization using a strong acidic solution—usually hydrochloric acid (HCl)—and is initiated with the addition of an oxidizing agent, such as ammonium persulfate (APS). At a sub-zero temperature, an increase in the molecular weight and crystallinity of PANI occurs [11,12,13]. 

The removal of impurities (such as byproducts related to doping, oxidant traces or oligomers of aniline) is imperative for the application of PANI in electronics or biomedical fields. The common strategies to purify PANI are centrifugation-based washing and dialysis. Centrifugation-based washing requires high centrifugal forces and several cycles of centrifugation, making it a tedious process and a not entirely effective method. The second strategy involves pouring PANI into a cavity surrounded by a dialysis membrane that takes several days to provide moderately pure PANI owing to small contaminants that pass through the membrane [14]. There are other proposed methods for the purification such as Soxhlet extraction [15] and precipitation–redispersion [16] that employ organic solvents. These purification methods of organic materials regularly require the solution of the material in an organic solvent. However, PANI is insoluble in a great number of organic solvents. Despite this, it has been shown that PANI emeraldine base (PANI-EB) has better solubility than polyaniline emeraldine salt (PANI-ES), which is the favored phase that is obtained from conventional synthesis of PANI. The transition between the different phases of PANI causes a strong color change from colorless to violet or from blue to green [15]. 

The aim of this work is to present results regarding the synthesis of PANI and PANI/AgNPs composite that optimizes oxidative polymerization. We also report a single phase of emeraldine base with a high percentage yield, for which a new purification process was implemented. 

## 2. Results and Discussion

### 2.1. PANI Characterization

#### 2.1.1. Fourier Transform Infrared Spectroscopy (FTIR)

Chemical characterization is shown in Figure 1. The comparison of FTIR spectra of the four products of PANI indicates that most of the characteristic infrared bands of the monomer are retained in the polymerized samples. The samples denominated as P1, P2, P3 and P4 refer to the products obtained by the four purification processes described in Section 3.3. PANI-P1 was obtained by the dispersion of a chloroform—methanol 10:1 solution and filtering. PANI-P2 was obtained by adding 1 M ammonium hydroxide. On the other hand, PANI-P3 resulted from the dispersion of a chloroform—methanol 10:1 system and heating the solution. PANI-P4 was obtained by adjustment of the pH value to 7.2 with the addition of ammonium hydroxide 1 M. All the processes were applied after phase separation with diethyl ether. The aniline spectrum shows absorption bands at 1621 and 1605 cm^−1^ regions, indicating ring deformation (Figure 1b). The bands at 1493 and 1461 cm^−1^ are due to ring stretching deformation. The stretching of the ring-N bond is observed at 1270 cm^−1^ and C–H out of plane deformation is detected at 743 and 687 cm^−1^ [17].

The FTIR spectra of PANI are in accordance with previously reported results [8,18,19,20], shown in Table 1. The main bands, corresponding to quinone and benzene ring stretching are observed around 1570 and 1486 cm^−1^, respectively. In addition, there are changes between the four products resulting from the purification processes. For instance, an increase in the band at 1300 cm^−1^ in PANI P2, PANI P3 and PANI P4 occurs, indicating a π-electron delocalization and the appearance of a peak at 1371 cm^−1^ corresponding to the C–N stretching vibrations near a quinonediimine unit, that are attributed to the PANI-EB phase [21].

#### 2.1.2. UV-Visible Spectroscopy (UV-Vis)

P1, P2, P3 and P4, see Figure 2, show different absorption bands that are reported in three of the PANI phases. In the PANI P1 spectrum, see Figure 2, green curve, two absorption bands are observed at 426 and 818 nm. These bands are reported in the PANI-ES phase, which has a green tonality and is poorly soluble in organic solvents [22,23]. The spectra of PANI P3, see Figure 2, violet curve, also has two absorption bands at 315 and 548 nm, the borders of which correspond to the violet-colored polyaniline pernigraniline base (PANI-PB) phase reported by Kang [2] and MacDiarmid [24]. 

On the other hand, the spectra of PANI P2, see Figure 2, blue curve, and PANI P4, see Figure 2, red curve, have absorption bands at similar wavelengths; at 320 and 569 nm in the first and 321 and 580 nm in the second curve. It has been reported by Nabid et al. [25] and Zhu et al. [26], that PANI-EB has absorption bands around 320 and 600 nm, which originate from the π→π* transition of the benzene rings and the extinction band of the quinoid ring, respectively. Therefore, it can be said that both products are PANI-EB. However, given that the solubility of PANI P4 appears to be better than PANI P2 (as observed in the variations obtained from the UV-Vis spectra), it was decided to continue experiments using PANI P4.

As shown in Figure 3, each phase of PANI can be identified by its distinct color, green for PANI-ES, see Figure 3a, blue for PANI-EB, see Figure 3b, and violet for PANI-PB, see Figure 3c. PANI clusters in the suspension form due to its low solubility, while in the PANI-EB and PANI-PB, the solubility is higher which can be observed by the color of the solvent.

#### 2.1.3. Scanning Electron Microscopy (SEM) and Electron Dispersive Energy (EDS)

The morphology found in the PANI was grain-like, with a globular sponge shape. Figure 4 shows the SEM images obtained for PANI P4 at 500× (Figure 4a) and 2000× (Figure 4b). A similar morphology was reported by Zhu, Peng and Jiang [26], where they explained that PANI has a coarse surface. Figure 4c presents the elemental analysis, which showed that mainly carbon was present and oxygen, sulfur and chlorine are detected in traces.

#### 2.1.4. Thermogravimetric Analysis and Differential Scanning Calorimetry (TGA-DSC)

The thermal study of PANI was performed using a simultaneous TGA-DSC analysis. The obtained TGA data are shown in Figure 5. The process of degradation was multistage, with three steps of decompositions. The first one, due to the moisture removal, was located around 100 °C, while the second one, observed between 250 °C and 300 °C, is attributed to the elimination of the low molecular weight oligomers and the dedoping process. Usually, this decomposition represents a greater weight loss; however, the purification step decreases the quantity of oligomers and PANI doped into 5% of the composition of the polymer. Additionally, thermogravimetric analysis has been reported with a weight loss in the range of 10–30% [26,27]. The third degradation stage is associated with the decomposition of PANI. On the other hand, the DSC analysis offers data by three endothermal fusions, the first one below 100 °C and the second one around 200 °C, which may be due to the removal of adsorbed water and low-weight molecules, respectively [28]. The third endothermal fusion, followed by major structural reorganization at 380 °C, is referred to as PANI degradation [29]. The product of PANI presents an exothermic curve due to energy liberation around 230.5 °C before structural reorganization occurs at 380 °C, the latter is probably due to its purification. The first derivative shows results consistent with the degradation process [30].

### 2.2. PANI/AgNPs Composite Characterization

#### 2.2.1. Fourier Transform Infrared Spectroscopy (FTIR)

Figure 6 shows the FTIR spectrum of PANI, as well as PANI/AgNPs composite synthesized in situ. The bands at 1570 and 1473 cm^−1^ are assigned to quinoid and benzenoid rings stretching vibrations, respectively, which also appear in the PANI/AgNPs composite spectrum at 1585 and 1499 cm^−1^.

The incorporation of AgNPs into PANI matrix causes a small shift of the bands towards higher wavenumbers, and the intensity of the peaks decreases, indicating interaction PANI-Ag. This effect was also demonstrated in previous work by Singh, Tiwari and Pandey [27]. The bands located at 400 cm^−1^ in the PANI/AgNPs spectra correspond to the presence of silver in the sample. The shift of the band from 1570 to 1585 cm^−1^ indicates that AgNPs may interact with nitrogen in the PANI within composite. This interaction was also presented by Gupta, Jana and Meikap [31].

#### 2.2.2. UV-Visible Spectroscopy (UV-Vis)

The PANI/AgNPs composite is shown in Figure 7. By means of this measurement, it can be defined that PANI used in PANI/AgNPs composite presents a similar behavior with PANI-EB by the bands located at 326 and 569 nm (Figure 7). In addition, it presents a third band at 450 nm due to the surface plasmon resonance (SPR) of AgNPs formed in PANI matrix. This was already reported by Nabid et al. [25] in composites chemically synthesized and in situ. Theorical calculations by MiePlot software were made to approximate the nanoparticle, where a SPR band at 450 nm is obtained for AgNPs with a diameter around 80 nm.

#### 2.2.3. Scanning Electron Microscopy (SEM) and Electron Dispersive Energy (EDS)

The images of PANI/AgNPs in situ composite at 500× (Figure 8a) and 2000× (Figure 8b) are observed, where it is presented a graininess morphology similar to PANI P4. However, it can be appreciated that PANI presents some areas with agglomeration. This behavior is due to the presence of silver in the matrix of PANI similar to the reported by Ran et al. [32], using gold instead of silver, where this morphology was detected when the gold precursor was added in a bigger amount. Also, in Figure 8c is presented the elemental analysis of PANI/Ag composite, where a silver signal is no detected, but the existing elements in PANI P4 are present.

#### 2.2.4. Thermogravimetric Analysis and Differential Scanning Calorimetry (TGA-DSC)

The thermal study includes a TGA-DSC simultaneous analysis (Figure 9). The TGA shows a multistage degradation process with three steps, similar to TGA of PANI. Meanwhile, the DSC analysis offer data of three exothermic fusions, the first previous to 100 °C assigned to water removal, followed by a second one around 250 °C due to low weight molecules removal, and a third one around 400 °C relative to polymer degradation [28,29]. The first derivative from differential thermogravimetric analysis (DTG) shows result consistent with the degradation process. The shifts between thermal analysis of pure PANI and PANI/AgNPs composites indicates the increased thermal stability of PANI when the AgNPs addition exists.

## 3. Materials and Methods

### 3.1. Materials

Silver nitrate (AgNO_3_, ≥99.0%), aniline monomer (C_6_H_5_NH_2_, ≥99.5%), ammonium persulfate ((NH_4_)_2_S_2_O_8_, 98%) and glycine (C_2_H_5_NO_2_, ≥98.5%) were obtained from Sigma-Aldrich (Toluca, Edo. Mex., México). Diethyl ether ((C_2_H_5_)_2_O, 99.90 %) was bought from CTR Scientific (Monterrey, N.L., México). Hydrochloric acid (HCl, 36.5–38%) and ammonium hydroxide (NH_4_OH, 28.7%) were obtained from J. T Baker (Phillipsburg, NJ, USA). Meanwhile, chloroform (CHCl_3_, 99.9%) was purchased from Fermont (Monterrey, N.L., México). Deuterated solvents including chloroform-*d* (CDCl_3_) and methanol-*d*_4_ (CD_3_OD) were used as received for NMR spectroscopic analysis. 

The FTIR spectra of the materials synthesized were acquired using a Thermo Scientific Nicolet spectrometer with a wavenumber between 4000 cm^−1^ and 400 cm^−1^ performing a total of 64 scans. The chemical structure of PANI was verified by ^1^H NMR using a 300 MHz Jeol spectrometer JNM-ECO300. The thermal stability, composition and purity of the materials were analyzed using TGA/DSC SDT-Q600 implementing a temperature program from 25 °C to 600 °C. A Jeol 5300 scanning electron microscope (SEM) with a dispersive energy detector (EDS) was utilized to observe the morphology of the samples and elemental analysis. UV-Vis spectra of PANI and composite were obtained by using a Jenway 6850 spectrophotometer in the wavelength range of 300 nm to 1000 nm. The samples deposited on glass substrates were analyzed with an uncoated glass substrate as a reference.

### 3.2. Polymerization of Aniline

In a beaker equipped with a magnetic stirrer and stirring rack, aniline monomer was acidified with 10 mL of HCl 1 M under constant stirring at 550 rpm by adding aniline dropwise at room temperature. Immediately after, HCl 1 M was poured until a pH close to zero was obtained. The mixture was placed in a cooling bath (which was prepared by a layer of pulverized ice in a crystallizer, followed by a layer of sodium chloride and another layer of pulverized ice) and stirred at 550 rpm. The mechanism, see Figure 10, involves an intricate interplay of consecutive chemical and physical reactions that need constant monitoring of the different included parameters in these reactions [18]. The pH and temperature monitoring were developed using an HI 2550 pH/ORP & EC/TDS/NaCl Meter Hanna Instruments. When reaching a temperature of −5 °C, 10 ml of APS 1.37 M ([APS]/[Aniline] molar ratio = 1.25) were added dropwise. Then, it was collocated under constant stirring for 2 h, maintaining the temperature at −5 °C during the entire reaction. 

### 3.3. Purification Process of PANI

In this step, it was recommended that a phase separation system where PANI doping was performed within the same procedure was used. This means obtaining a higher yield percentage from the PANI-EB than was previously reported. The product obtained in the aniline polymerization was treated in an extraction funnel with 30 mL of diethyl ether and 23 ml of 1 M ammonium hydroxide, which was stirred and depressurized. This allowed a phase separation by precipitation of the organic phase. In addition, this enabled a pH change in the PANI which is in its PANI-ES phase with an acidic pH after the polymerization.

A portion of organic solvent with brown contaminants was removed from the extraction funnel with the help of a Pasteur pipette, while the aqueous phase was filtered using Whatman #43 filter paper. The bulk product was kept in the filter while the precursors and/or unreacted by-products were removed by filtration. Then, four different processes were carried out to continue the purification. This stage is important because they can negatively influence the subsequent properties. Furthermore, to ensure its application in the biomedical field, cytotoxicity in biological compounds is associated with components of low molecular weight that must be eliminated from the material [15]. The main problem in the electronic applications for this compound are the impurities surrounding the AgNPs over PANI, preventing the current from passing through the two materials correctly.

#### 3.3.1. Process 1

The product of the filter paper, obtained in the previous filtrate, was dispersed in 500 mL of chloroform–methanol in a volumetric ratio of 1:1 by magnetic stirring at room temperature for 10 min and refiltered using a filter paper, Whatman #43. The resulting product by this process did not have a great solubility in organic solvents, its coloration was emerald green and a pH value of 2 was acquired. It was filtered again and put into a vacuum oven for drying by approximately 24 h, resulting in PANI P1.

#### 3.3.2. Process 2

The reaction system was modulated according to pH by employment of ammonium hydroxide 1 M. It was increased to a pH value of 10 from the stage of the extraction funnel, adding more ammonium hydroxide, which caused the phase change to PANI-EB. This product (PANI P2) was more soluble in solvents; however, PANI was lost in the filtrate due to this reason.

#### 3.3.3. Process 3

The product obtained in the filtrate before the polymerization was dispersed in 500 mL of chloroform–methanol in a volumetric ratio of 1:1 by magnetic stirring at room temperature for 10 min. Then, the temperature was raised to 40 °C in the PANI suspension with chloroform–methanol to try to increase its solubility. This caused a change in the coloration from green to violet, characteristic of the PANI-PB. The suspension was filtered using a filter paper, Whatman #43. The filtration allowed a seemingly black powder to be obtained that was dried in the vacuum oven as in the previous processes used for obtaining PANI P3.

#### 3.3.4. Process 4

After the first filtration was developed, a wash with deionized water was carried out reaching a pH value of 2. It was filtered, and1 M ammonium hydroxide drops were added until a pH value of 7.2 was reached. It was subsequently filtered with filter paper, Whatman #43, and washed again with deionized water and drying at 30 Pa. The product obtained was a bluish powder (PANI P4) with 72% yield. ^1^H NMR (300 MHz, CDCl_3_ and CH_3_OH): δ = 2 ppm (s, J = 982 Hz, CH_3_(CO)CH_3_), δ = 2.87 ppm (s, J = 1435 Hz, shielded N–H), δ = 2.85 ppm (s, J = 1656 Hz, shielded Ar–H), δ = 4.6–5.0 ppm (m, J = 2296 Hz, heterojunction structures placed by oxidation and CH_3_OH), δ = 7.9 ppm (s, J = 3952 Hz, shielded CDCl_3_), see Figure 11.

### 3.4. Synthesis of PANI/AgNPs in Situ Composite

The composite, see Figure 12, was carried out in a beaker equipped with a magnetic stirrer by direct addition of 1 mL of silver nitrate 3 mM and 10 µl of glycine 0.6 M in a molar ratio of [Gly]/[AgNO_3_] = 2 into the mixture of aniline acidified to pH ≈ 0 with HCl 1 M. The suspension was stirred at 550 rpm for 10 min and then collocated into the cooling bath, where 10 mL of APS 1.37 M was added at −5 °C. Afterwards, the stirring continued for an additional 2 h under the same conditions. The bulk product was then purified by the proposed purification for the process 4. The final product was a bluish powder. 

The most feasible approach to the synthesis and stabilization mechanism of AgNPs inside a PANI matrix is by the oxidation of aniline and posterior reduction of silver ions to Ag^0^. Limitation of aggregation is observed for PANI itself where amine groups stabilize AgNPs, see Figure 13.

### 3.5. Theorical Prediction of Surface Plasmon Resonance

The theoretical prediction of the peak due to Surface Plasmon Resonance (SPR) of AgNPs embedded in PANI was made by Mie calculations, using the software “MiePlot v4612”. The particle size distribution was theoretically calculated by plotting the extinction, scattering and absorption cross-sections versus the wavelength. The sphere was selected as silver and the surrounding medium was selected as water. The wavelength selected was from 200 to 600 nm. The corresponding data obtained from the model were compared to UV-Vis results from the experiment.

## 4. Conclusions

PANI was successfully prepared via oxidative polymerization at the molar ratio of the oxidant to aniline of 1.25, controlling pH and temperature, and it was characterized for its chemical, optical, thermal and microstructural properties. The purification of PANI is crucial for its sensing application. Therefore, a new method for its purification was applied based on phase separation by organic solvents, finding a way to dedope PANI during the process, and prevent it from undergoing a phase change. The reduction of the contaminant was observed by TGA for both materials, verified by the decrease in weight loss in comparison to the literature for the impurities characteristic of the synthesis. PANI/AgNPs in in situ composite were also synthesized using APS as the oxidant, AgNO_3_ as the co-oxidant and silver precursor. This was verified by chemical and optical characterization. It was found that the morphology of these materials was a grainy shape where silver agglomerates PANI around itself in the composite. Finally, the above implies the stabilization mechanism of AgNPs (Ag^0^) by amino groups of PANI. As future work, the enzyme immobilization development on PANI and PANI/AgNPs activated materials is proposed, in order to apply them successfully and improve the functionality of biosensors.

## Figures and Tables

**Figure 1 molecules-24-01621-f001:**
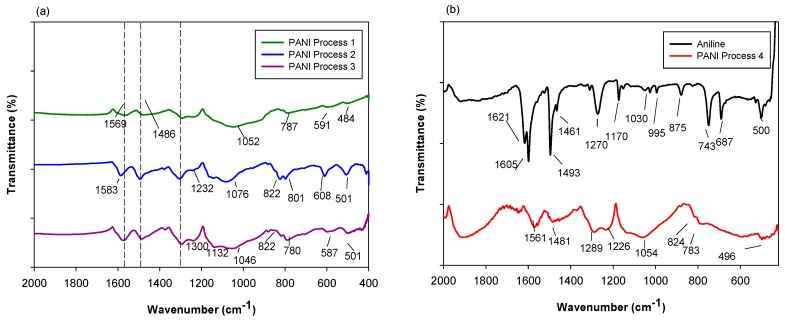
(**a**) Fourier transform infrared (FTIR) spectra of polyaniline (PANIs) resulting from the purification processes 1, 2 and 3; (**b**) FTIR spectra of aniline and PANI resulting from the purification process 4.

**Figure 2 molecules-24-01621-f002:**
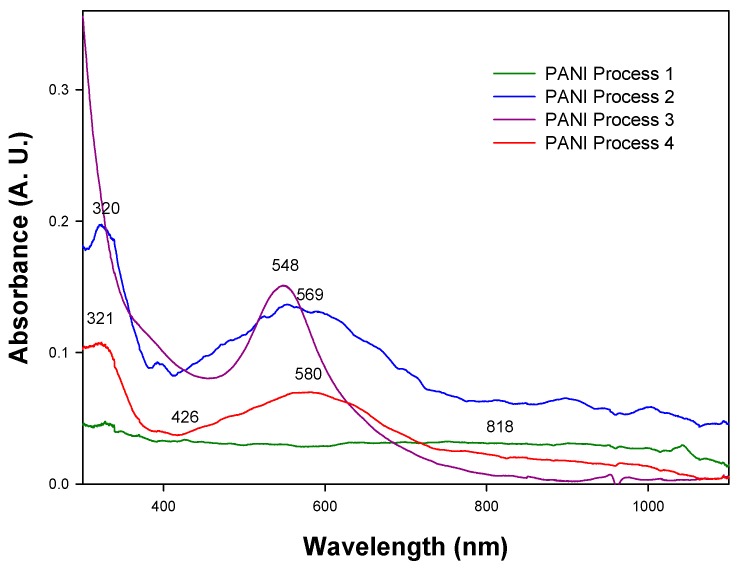
UV-Vis spectra of the PANIs resulting from the purification processes 1, 2, 3 and 4.

**Figure 3 molecules-24-01621-f003:**
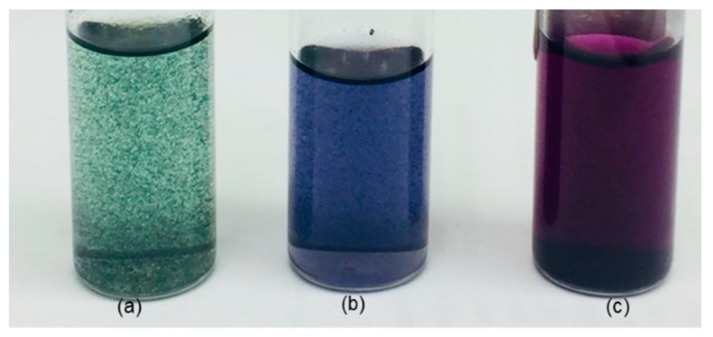
Color identification of the phases of PANI obtained with the purification processes: (**a**) polyaniline emeraldine salt (PANI-ES), (**b**) polyaniline emeraldine base (PANI-EB) and (**c**) polyaniline pernigraniline base (PANI-PB).

**Figure 4 molecules-24-01621-f004:**
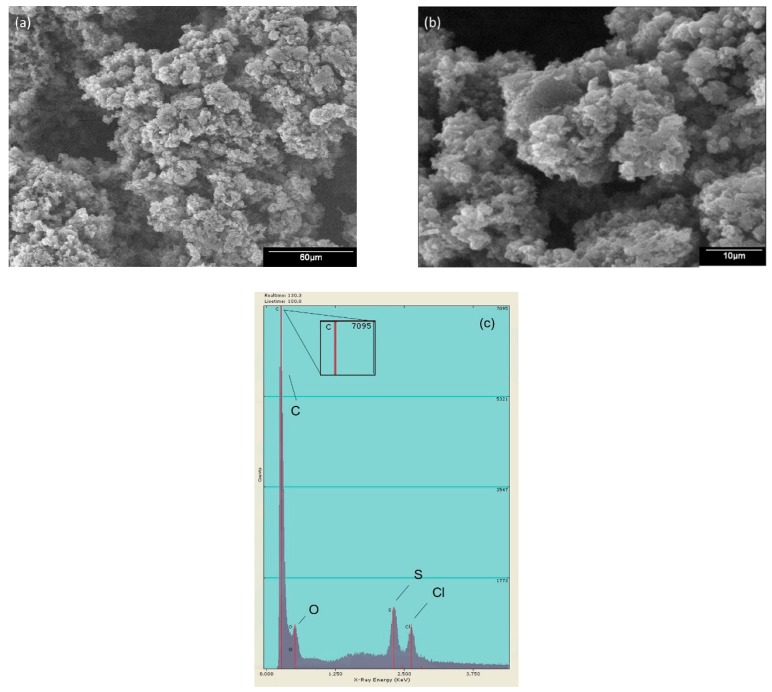
SEM images of PANI-P4 magnified to (**a**) 500× and (**b**) 2000× and (**c**) EDS analysis.

**Figure 5 molecules-24-01621-f005:**
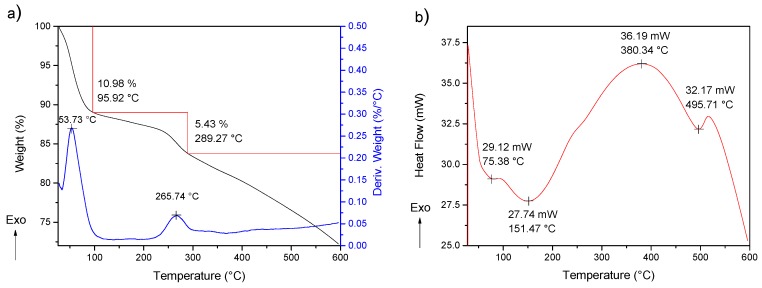
(**a**) TGA-DTG and (**b**) DSC of PANI-P4.

**Figure 6 molecules-24-01621-f006:**
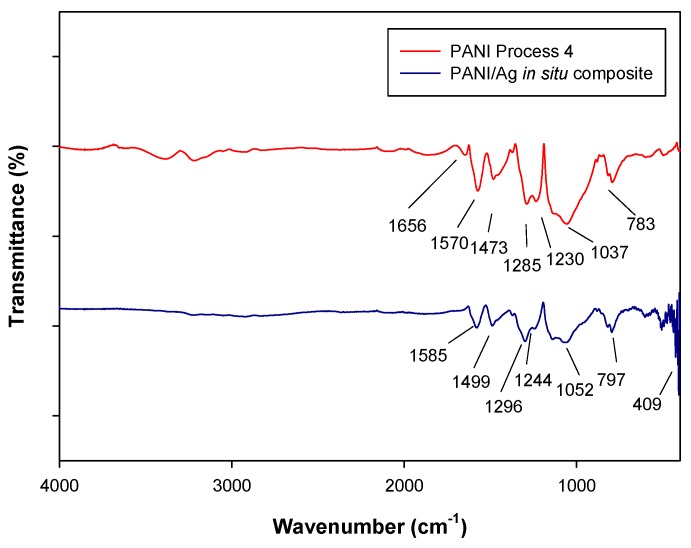
FTIR spectra of PANI-P4 and PANI/AgNPs in situ composite.

**Figure 7 molecules-24-01621-f007:**
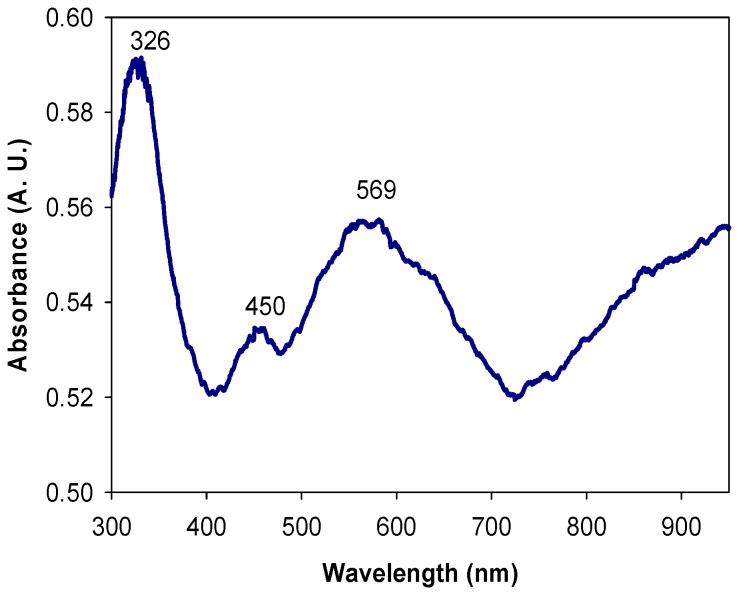
UV-Vis spectra of PANI/AgNPs in situ composite.

**Figure 8 molecules-24-01621-f008:**
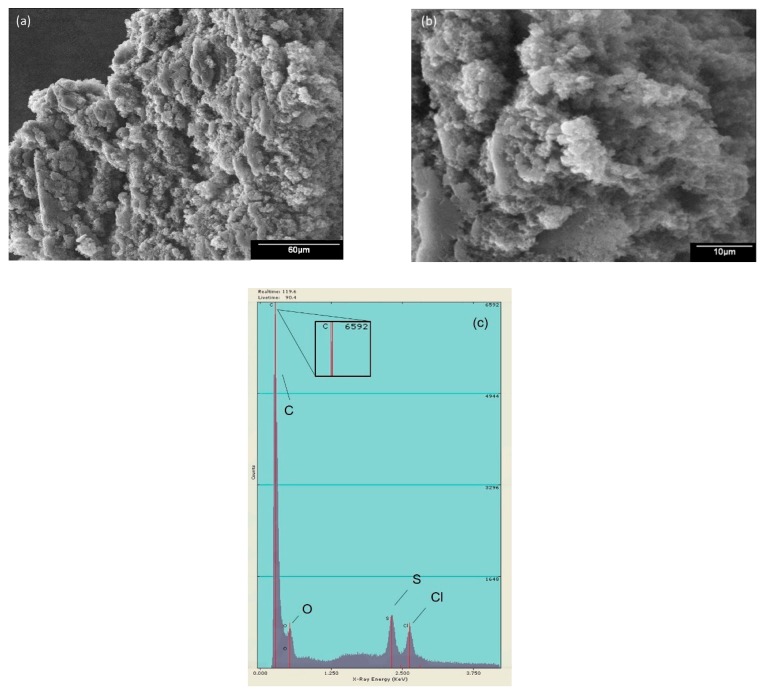
SEM images of PANI/AgNPs in situ composite magnified to (**a**) 500× and (**b**) 2000× and (**c**) Electron Dispersive Energy (EDS) analysis.

**Figure 9 molecules-24-01621-f009:**
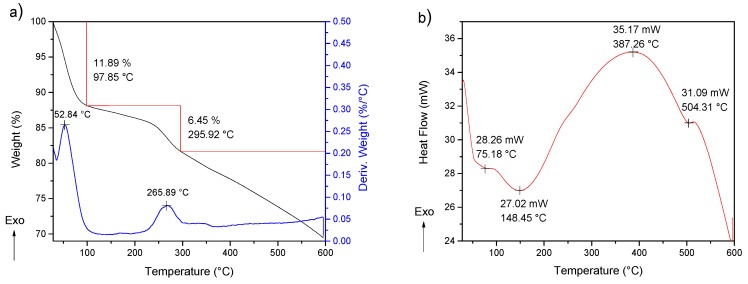
(**a**) TGA-DTG and (**b**) DSC of PANI/AgNPs in situ composite.

**Figure 10 molecules-24-01621-f010:**
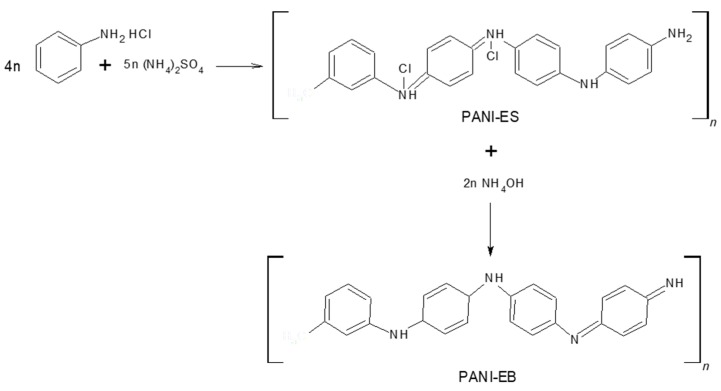
Synthesis route of PANI-ES and PANI-EB.

**Figure 11 molecules-24-01621-f011:**
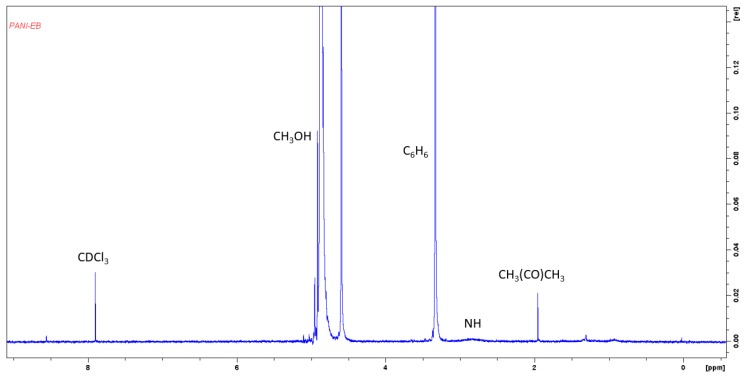
NMR spectra of PANI resulting from the purification process 4.

**Figure 12 molecules-24-01621-f012:**
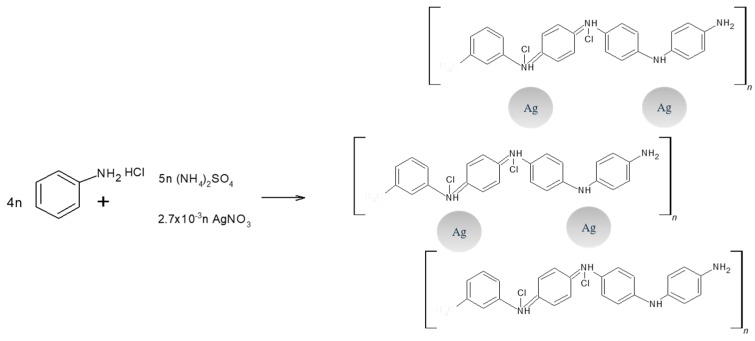
Synthesis route of PANI/AgNPs composite.

**Figure 13 molecules-24-01621-f013:**
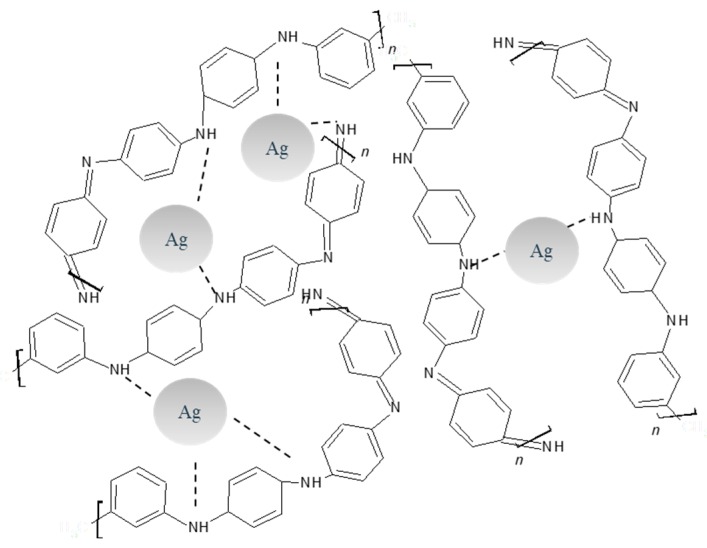
Stabilization mechanism of PANI/AgNPs composite.

**Table 1 molecules-24-01621-t001:** Assignment of PANI bands as identified elsewhere [2,9,18,19,20,21,22].

PANI P1 (cm^−1^)	PANI P2 (cm^−1^)	PANI P3 (cm^−1^)	PANI P4 (cm^−1^)	Assignment
1569	1583	1569	1561	Quinoid ring stretching
1486	1486	1486	1481	Benzenoid ring stretching
-	1378	1373	1371	C–N stretching vibration near quinonediimine unit
1300	1300	1300	1289	C–N stretching in *cis*-Q-B-Q ^1^, Q-B-B ^2^ and B-B-Q ^3^
1232	1232	1232	1226	C–N stretching in B-B-B ^4^
1052	1076	1046	1054	
-	822	822	824	C–H out of plane bending of 1,2,4-ring
787	801	780	783	
591	608	587	-	
484	501	501	496	

^1^ Q-B-Q quinoid unit-benzenoid unit-quinoid unit. ^2^ Q-B-B quinoid unit-benzenoid unit-benzenoid unit. ^3^ B-B-Q benzenoid unit-benzenoid unit-quinoid unit. ^4^ B-B-B benzenoid unit-benzenoid unit-benzenoid unit.

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
