# Peer review of "Synthesis and Novel Purification Process of PANI and PANI/AgNPs Composite"

_molecules, 2019, doi:10.3390/molecules24081621_

Round 1
Reviewer 1 Report
Mota et al. presented a series of novel purification processes of PANI to increase the percentage yield of PANI synthesis. In addition, the PANI/AgNPs composite was prepared during the polymerization of aniline in situ. The experimental results were organized systematically and the manuscript was well-written. I recommend this manuscript can be published in Molecules after a major revision. Followings are the remarks for the author:
1. Although the author described the different between P1-P4 in Experimental section, it will be good if the author can have a short highlight of each PANI purified using different method in Section 2.1, which can make the comparison become clearer.
2. The TGA results are weird. Although the weight decay can be explained, usually it should not have such big degrade at the range of 100-200 oC, as published in literature (Synthetic Metal, 1989, 30, 321-325). The authors should carefully purified and remove solvents from the system and measure the studied PANIs again.
3. For the PANI/Ag composite, no Ag signal was detected by EDS analysis using SEM. This may due to the morphology of PANI/Ag composite, and a core (Ag)-shell (PANI) structure is expected. The authors should characterize the PANI/Ag composite using TEM instead of SEM, and see if the EDS analysis of TEM can detect Ag or not.
4. Lots of grammar and typo errors are found in the manuscript, the authors should take care of those errors.
Author Response
Response to Reviewer 1 Comments
Dear Reviewers:
We sincerely thank the reviewers for the time invested in reviewing our paper as well as their valuable comments. We have revised the paper taking into account the comments provided. We are now submitting the revised manuscript for review and potential publication.
Response to reviewer #1
General comments
Point 1) Although the author described the different between P1-P4 in Experimental section, it will be good if the author can have a short highlight of each PANI purified using different method in Section 2.1, which can make comparison become clearer.
Author response: Short description of each PANI was added to manuscript in Section 2.1.
Point 2) The TGA results are weird. Although the weight decay can be explained, usually it should not have such big degrade at the range of 100-200 °C, as published in literature (Synthetic Metal, 1989, 30, 321-325). The authors should carefully purified and remove solvents from the system and measure the studied PANIs again.
Author response: After the comparison with some references, it was not observed big difference with the TGA analysis. However, we opted for separating graphs and labeling every graph to discard possible confusion between analyses.
Point 3) For the PANI/Ag composite, no Ag signal was detected by EDS analysis using SEM. This may due to the morphology of PANI/Ag composite, and a core (Ag)-shell (PANI) structure is expected. The authors should characterize PANI/Ag composite using TEM instead of SEM, and see if the EDS analysis of TEM can detect Ag or not.
Author response: Although silver is not detected by EDS analysis due to the morphology obtained, silver presence is corroborated by FTIR analysis and UV-Vis spectra where SPR of spherical AgNPs is clearly reported. There are some works published where AgNPs formation is reported by the same characterizations (for example,Nabid, M.R.; Asadi, S.; Sedghi, R.; Bayandori Moghaddam, A. Chemical and enzymatic polymerization of polyaniline/ag nanocomposites. Chemical Engineering and Technology 2013, 36, 1411–1416; Bober, P.; Stejskal, J.; Trchová, M.; Prokeš, J. In-situ prepared polyaniline-silver composites: Single- and two-step strategies. Electrochimica Acta 2014, 122, 259–266)
Point 4) Lots of grammar and typo errors are found in the manuscript, the authors should take care of those errors.
Author response: We have reviewed the manuscript carefully and made changes where it was considered necessary. We believe the improved grammar and punctuation will better serve the reader.

Reviewer 2 Report
Molecules
Manuscript ID: molecules-487100
Title: Synthesis and novel purification process of PANI and PANI/AgNPs composite.
Reviewer comments:
The authors have produced an interesting study concerning the synthesis of new Polyaniline-based composites, using silver as doping agent. The work has been undertaken diligently, it’s written in a clear way and the interpretation of the data supports the conclusions. However, some points should be modified and investigated, for example structural and conductive analyses are missing. I would suggest publishing it on Molecules only after major revisions.
(1) Introduction: The authors detail several examples about the use of Polyaniline in the field of organic electronics and show new processes related to the synthesis and purification of PANI composites, but they don’t discuss the conductive properties of the material obtained. The point is: the conductive properties of PANI composites obtained via the new synthetic routes are comparable or not with the ones of PANI synthesized via common synthetic routes?
(2) FT-IR spectra: The baseline of FT-IR spectra has a quite strange behavior because it is characterized by a relevant decrease in terms of transmittance…the authors should improve the quality of FT-IR analyses correcting the baseline behavior.
(3) Figure 3: PANI-ES and PANI-EB samples seem to be suspensions not solutions
(4) EDS analyses: The authors should improve the quality of all EDS analyses…
(5) Materials and Methods: The section about 1H NMR spectroscopy is not clear because the deuterated solvents indicated are not detectable in the corresponding spectra.
(6) General remark: The structure of all PANI and PANI/Ag composites can be studied via XRD analyses; furthermore, this technique can highlighted the effect of dopant on PANI structure. I think that the authors should investigate this point as similar works reported in literature (for example Sanches, E. A., et al. "Structural characterization of Chloride Salt of conducting polyaniline obtained by XRD, SAXD, SAXS and SEM." Journal of Molecular Structure 1036 (2013): 121-126; Raghunathan, Anasuya, G. Rangarajan, and D. C. Trivedi. "13C CPMAS NMR, XRD, dc and ac electrical conductivity of aromatic acids doped polyaniline." Synthetic metals 81.1 (1996): 39-47).
Author Response
Response to Reviewer 2 Comments
Dear Reviewers:
We sincerely thank the reviewers for the time invested in reviewing our paper as well as their valuable comments. We have revised the paper taking into account the comments provided. We are now submitting the revised manuscript for review and potential publication.
Response to reviewer #2
General comments
Point 1. Introduction: The authors detail several examples about the use of Polyaniline in the field of organic electronics and show new processes related to the synthesis and purification of PANI composites, but they don’t discuss the conductive properties of the material obtained. The point is: the conductive properties of PANI composites obtained via the new synthetic routes are comparable or not with the ones of PANI synthesized via common synthetic routes?
Author response: It was decided to simply focus on optical and chemical applications of PANI whose properties have been studied in this work. However, it is not discarded the study of conductive properties as future work.
Point 2. FT-IR spectra: The baseline of FT-IR spectra has a quite strange behavior because it is characterized by a relevant decrease in terms of transmittance… the authors should improve the quality of FT-IR analyses correcting the baseline behavior.
Author response: Correction of baseline in all FTIR spectrums was added to the manuscript.
Point 3. Figure 3: PANI-ES and PANI-EB samples seem to be suspensions not solutions.
Author response: Indeed, they were suspensions. The term was corrected in the manuscript.
Point 4. EDS analyses: The authors should improve the quality of all EDS analyses…
Author response: The quality of EDS analyses was improved.
Point 5. Materials and Methods: The section about 1H NMR spectroscopy is not clear because the deuterated solvents indicated are not detectable in the corresponding spectra.
Author response: Discussion about the solvents and labels for indication of each peak were added to the manuscript.
Point 6. General remark: The structure of all PANI and PANI/Ag composites can be studied via XRD analyses; furthermore, this technique can highlighted the effect of dopant on PANI structure. I think that the authors should investigate this point as similar works reported in literature (for example Sanches, E. A., et al. “Structural characterization of Chloride Salt of conducting polyaniline obtained by XRD, SAXD, SAXS and SEM.”
Author response: We accept XRD analyses can provide further information of PANI/Ag interactions, however, for terms of the synthesis, which is what is reported, the data provided by FTIR and UV-Vis characterizations corroborate the chemical interaction between PANI/Ag also reported by other published works (for example, Nabid, M.R.; Asadi, S.; Sedghi, R.; Bayandori Moghaddam, A. Chemical and enzymatic polymerization of polyaniline/ag nanocomposites. Chemical Engineering and Technology 2013, 36, 1411–1416; Bober, P.; Stejskal, J.; Trchová, M.; Prokeš, J. In-situ prepared polyaniline-silver composites: Single- and two-step strategies. Electrochimica Acta 2014, 122, 259–266)

Round 2
Reviewer 1 Report
The manuscript is improved after revision. However, the reviewer still feels the characterization of Ag is needed. The author should as least try to use TEM to investigate their system. A major revision is recommended.
Reviewer 2 Report
-